# Parasite Occurrence and Parasite Management in Swedish Horses Presenting with Gastrointestinal Disease—A Case–Control Study

**DOI:** 10.3390/ani10040638

**Published:** 2020-04-07

**Authors:** Ylva Hedberg-Alm, Johanna Penell, Miia Riihimäki, Eva Osterman-Lind, Martin K. Nielsen, Eva Tydén

**Affiliations:** 1Horse Clinic, University Animal Hospital, Swedish University of Agricultural Sciences, 750 07 Uppsala, Sweden; 2Division of Veterinary Nursing, Department of Clinical Sciences, Swedish University of Agricultural Sciences, 750 07 Uppsala, Sweden; Johanna.Penell@slu.se; 3Equine Medicine Unit, Department of Clinical Sciences, Swedish University of Agricultural Sciences, 750 07 Uppsala, Sweden; Miia.Riihimaki@slu.se; 4National Veterinary Institute, Department of Microbiology, Section for Parasitology diagnostics, 751 89 Uppsala, Sweden; eva.osterman-lind@sva.se; 5Maxwell H. Gluck Equine Research Center, Department of Veterinary Science, University of Kentucky, Lexington, KY 40546, USA; martin.nielsen@uky.edu; 6Parasitology Unit, Department of Biomedical Science and Veterinary Public Health, Swedish University of Agricultural Sciences, 750 07 Uppsala, Sweden; eva.tyden@slu.se

**Keywords:** horse, colic, gastrointestinal disease, *Strongylus vulgaris*, *Anoplocephala perfoliata*, cyathostominae

## Abstract

**Simple Summary:**

Abdominal pain, colic, is a common clinical sign in horses, sometimes reflecting life-threatening disease. One cause of colic is parasitic infection of the gut. Various drugs, anthelmintics, can be used to reduce or eliminate such parasites. However, frequent use has led to problems of drug resistance, whereby many countries now allow anthelmintics to be used on a prescription-only basis. In Sweden, this has led to a concern that parasitic-related colic in horses is increasing. This study aimed to investigate whether horses with colic differed in parasitological status compared to horses without colic. A secondary aim was to collect information regarding current parasite control measures used by horse owners. Exposure to *S. vulgaris*, a parasite with the potential to cause life-threatening disease, appeared high as determined by the presence of antibodies in the blood. Horses with inflammation in the abdominal cavity had higher antibody levels than other causes of colic. Despite new legislation, 29% of owners did not use fecal analyses for parasites and the use of extended methods to diagnose specific parasites was low. Also, owners rarely used alternative methods to reduce the pasture parasite burden. The study suggests a need for education in the use of both fecal analyses and pasture management.

**Abstract:**

All grazing horses are exposed to intestinal parasites, which have the potential to cause gastrointestinal disease. In Sweden, there is a concern about an increase in parasite-related equine gastrointestinal disease, in particular *Strongylus vulgaris*, since the implementation of prescription-only anthelmintics approximately 10 years ago. In a prospective case–control study, parasitological status, using fecal analyses for strongyle egg counts, the presence of *Anoplocephala perfoliata* eggs and *S. vulgaris* Polymerase chain reaction (PCR) as well as serology for *S. vulgaris*, were compared between horses presenting with or without gastrointestinal disease at a University hospital during a one-year period. Information regarding anthelmintic routines and pasture management was gathered with an owner-filled questionnaire. Although the prevalence of *S. vulgaris* PCR was 5.5%, 62% of horses were positive in the enzyme-linked immunosorbent assay (ELISA) test and horses with peritonitis showed higher antibody levels for *S. vulgaris*, as compared to other diagnoses or controls. Overall, 36% of the horse owners used only fecal egg counts (FEC), 32% used FEC combined with specific diagnostics for *S. vulgaris* or *A. perfoliata,* and 29% dewormed routinely without prior parasite diagnostics. Effective management methods to reduce the parasitic burden on pastures were rare and considering exposure to *S. vulgaris* appears high; the study indicates a need for education in specific fecal diagnostics and pasture management.

## 1. Introduction

Colic in horses is a significant cause of mortality and morbidity and thus, is a major contributing factor to economic loss within the equine industry [1]. A vast number of potential risk factors have been implicated as a cause of gastrointestinal disease in horses, including intestinal parasites [2,3]. As grazing animals, horses are hosts to a large number of internal parasites that, to a varying degree, may result in pathology of the gastrointestinal tract [2].

Of all the equine internal parasites, *Strongylus vulgaris* is regarded as the most pathogenic and was therefore the main target of parasitic control programs in the 1960s, when modern anthelmintic drugs were first introduced [4]. This parasite has a long lifecycle of about six months, of which four months involve migration of larvae in the mesenteric arteries, in particular the cranial mesenteric artery [5]. Due to its larval migration within the mesenteric vasculature, *S. vulgaris* may cause severe arterial inflammation and damage to the otherwise smooth endothelial surface, with subsequent thrombus formation [5]. This potentially leads to the occlusion of arteries and arterioles supplying the intestinal wall, causing intestinal infarction and septic peritonitis. This condition is often fatal and requires surgical resection of the infarcted intestinal segment for a chance of survival [6].

The cestode *Anoplocephala perfoliata* has also been implicated as a cause of colic of varying degrees of severity [7]. This parasite’s predilection for the ileocecal junction may cause lesions in this area, such as ileocecal invaginations and ileal impactions, most likely due to parasite-induced intestinal pathology at the site of attachment, but perhaps also secondary to physical obstruction and/or modifications in intestinal motility [8,9]. In addition, studies have demonstrated an association between *A. perfoliata* infection and spasmodic colic [10] as well as colic in general [11].

In contrast to other parasites, small strongyles, cyathostomins, have a direct life cycle and the early third larval stage can be arrested in the gut wall as encysted larvae for prolonged periods of time. The emergence of large numbers of such arrested larval stages can cause a severe inflammatory reaction in the colon and cecum resulting in diarrhea, sometimes with fatal outcomes, predominately in young animals; a condition known as larval cyathostominosis [12,13]. Moreover, large burdens of cyathostomins have been related to ill-thrift and weight-loss [14,15]. However, it is still under debate if cyathostomins are associated with colic in general, and a recent case–control study was unable to demonstrate a connection between cyathostomins and colic [16]. 

In the 1960s, regular anthelmintic treatment of horses was introduced, and as a result, some previously common parasite-induced gastrointestinal diseases, such as non-strangulating intestinal infarction caused by *S. vulgaris*, appeared to be largely eradicated [2]. However, the increasing reports of anthelmintic resistance in both cyathostomin parasites and *Parascaris* spp. [17,18], have led several countries, including Denmark, Finland, the Netherlands, Italy and Sweden, to implement a prescription-only sale of equine and livestock anthelmintic drugs [19]. In Sweden, the legislation advocates the prescription of anthelmintics to adult horses only after fecal analyses. By treating horses exceeding a chosen cut-off value, often 200 strongyle eggs per gram of feces (EPG), fewer anthelmintic treatments are administered, thus reducing the risk of developing of anthelmintic resistance [19,20]. However, since the implementation of more selective anthelmintic treatment strategies, Denmark and Sweden have recently documented a clear increase in the prevalence of *S. vulgaris* in the equine population in these countries [21,22]. Furthermore, post-mortem studies conducted in Italy indicate that pathological lesions caused by *S. vulgaris* are still widespread and in Denmark, several clinical cases of *S. vulgaris*-associated non-strangulating infarctions with subsequent septic peritonitis with a high mortality rate have been reported [6,23,24]. 

In Sweden, the legislation restricting anthelmintic products to prescription-only came into effect in October 2007. Just prior to this, only a minority (1%) of Swedish horse owners based anthelmintic treatment on fecal analyses of their horses [25]. As yet, there are few studies investigating how the prescription-only legislation has affected anthelmintic routines in Sweden and its possible effect on parasite-related disease in Swedish horses. Despite similar legislation in several European countries, there appears to be substantial national differences in how the law is implemented in practice [19]. Recently, an *S. vulgaris* prevalence of 61% was demonstrated in Swedish horse farms [22]. Whether this has resulted in an increase in clinical disease in Swedish horses is, however, unknown.

The aim of the present study was to compare the parasitological status between horses presenting for gastrointestinal disease (colic, colitis, peritonitis, weight loss) and horses presenting for non-intestinal disease at a university equine referral hospital during a one-year period, using coprological and serological assays. In addition, a secondary aim was to gather information regarding current anthelmintic treatment routines and pasture management practices using an owner-filled questionnaire at admission.

## 2. Materials and Methods 

The study was approved by the Uppsala animal welfare ethics committee (license number: Dnr 68/16). Owner consent was required for study participation and participating owners were asked to fill in a consent form at admission.

### 2.1. Project Outline

The study was designed as a prospective case–control study at the Horse Clinic, University Animal Hospital, Swedish University of Agricultural Sciences, over a one-year period (February 2017–February 2018). The Horse Clinic receives both first opinion as well as referred equine cases with an annual case load of approximately 6000 cases. A horse presenting to the clinic and diagnosed with a disease related to the gastrointestinal canal was classified as a case. Each case was matched with a control horse presenting during the same week and, when possible, of same or similar age, and diagnosed with either a disease unrelated to the gastrointestinal canal (e.g., respiratory, skin, eye, cardiac, and orthopedic conditions) or presenting for prophylactic reasons (e.g., general health check, oral examination). Fecal and blood samples were collected by veterinarians from each case and control during the horse´s visit at the clinic. In addition, each owner was asked to respond to a questionnaire for collecting information regarding previous colic history, previous anthelmintic treatment, anthelmintic routines, and pasture management (Table 1). Regarding the question about deworming routines, there were five alternatives for the horse owner with the highest number of treatments being 2–4 times/year (Table 1). The reason for this is that in Sweden the recommendation for adult horses is to perform parasite diagnostics in April/May and only treat individuals exceeding 200 EPG and positive for *S. vulgaris* and/or *A. perfoliata* [26]. It is very rare that adult horses are dewormed more than 4 times/year in Sweden. 

### 2.2. Processing of Samples

Each horse included in the study had a fecal sample collected either at the time of rectal exam (majority of case horses), or from feces passed in the loose box (cases and controls) or transport vehicle in which the horse was brought to the clinic (controls). Fecal samples, approximately 200 g, were stored at 4 °C until parasite analyses the following day. In addition, two serum blood collection tubes (10 mL BD Vacutainer® (BD Technologies, Durham, NC, USA)) were obtained, either by vacutainer technique (20 G × 25 mm) from the jugular vein, or, in some case horses, from a permanent jugular catheter (Extended Use MILACATH^®^ (MILA International Inc, Florence, KY, USA) placed in the jugular vein. In the latter case, at least 10 mL of blood was discarded before obtaining the sample. Blood samples were refrigerated at 4 °C for 12–24 h and then centrifuged at 2000× g for ten minutes. Serum was pipetted off and stored in 1.8 mL cryo tubes at −80 °C until the enzyme-linked immunosorbent assay (ELISA) was performed. 

### 2.3. Parasite Analyses

Strongyle fecal egg counts (FECs) were carried out for each horse using a modified McMaster technique with a minimum detection limit of 50 EPG [27]. Parasite eggs in fecal samples (3 g) were floated using a saturated NaCl solution (SG = 1.18) [27]. Irrespective of FEC, larval cultures for detection of *S. vulgaris* were performed on 50 g feces from each horse according to Bellaw and Nielsen [28]. In brief, feces were mixed with an equal volume of vermiculite (Weibulls, Sweden), tap water was added to obtain a moist condition and samples were cultured at 20 °C for 14 ds. Third stage larvae were harvested after sedimentation for 12–16 h at 20 °C by the inverted Petri dish method [29]. Pellet of harvested L3 was performed by collecting approximately 20 mL of the fluid into a 50 mL Falcon tube, centrifuging at 248× g for three minutes and discarding the supernatant. Larval DNA was extracted with NuceloSpin^®^Tissue (Macherey-Nagel, Düren, Germany) according to the manufacturer´s instruction. An ITS-2 specific PCR was used for identification of *S. vulgaris* according to Nielsen et al. (2008) and Tydén et al. (2019) [22,30].

*Anoplocephala perfoliata* were examined on 30 g feces using the modified flotation technique described by Berozoa et al. [31]. Feces were mixed with 60 mL tap water, sieved and collected into four 15 mL test tubes and centrifuged 1000× g for ten minutes. The supernatant was removed and pellet suspended in sugar salt solution (saturated sodium chloride solution with 50% glucose with a density of 1.280) to form a convex meniscus. Cover glasses (18 mm × 18 mm) were put on the top and centrifuged for 5 min at 214× *g* (no brakes) in a swing out centrifuge. Samples were left 5 min after centrifugation and thereafter transferred to microscope slides and microscopically examined at 40–100× magnification.

### 2.4. Strongylus vulgaris Specific ELISA

Serum samples were packed with dry ice and shipped to the Nielsen Parasitology Laboratory at the University of Kentucky. Here, an indirect ELISA using recombinant SvSXP protein as antigen was performed as previously described [32]. Samples were diluted 1:50 and horseradish peroxidase-conjugated goat anti-horse IgG(T) (Bethyl Laboratories, Inc., Montgomery, TX, USA) was used as a secondary antibody at a dilution of 1:40,000. Results were reported as the normalised value, percentage of a positive control to reduce inter-assay variability [32]. The positive control sample was obtained from a horse known to be infected with *S. vulgaris*. 

### 2.5. Statistical Analyses

All statistical analyses were interpreted as statistically significant up to the *p*-value  <  0.05 with the application of Bonferroni correction of the *p*-value, i.e., the division of the *p*-value for significance with the number of tests applied, to account for multiple testing, when appropriate. All statistical analyses were performed using Stata SE (version 14.2, StataCorp, College Station, TX, USA).

#### 2.5.1. Data on Horses, Colic Diagnoses and Season

Statistical analyses on data on horses, colic diagnosis and season were performed as follows. Horses were categorised into six age-groups (1–2; 3–5; 6–10; 11–15; 16–20 and >20 years) and gender (mare, stallion, gelding). Gastrointestinal disease was classified into nine categories: impaction (large colon, small colon, and cecal), large intestinal displacement, acute colitis, peritonitis, equine eosinophilic gastroenteritis/enterocolitis, gas distention colic, chronic colitis, undiagnosed, and other. Season was categorised into four seasons (December–February, March–May, June–August, September–November). 

Comparisons between diagnostic categories and season was performed using Fisher’s exact test. Comparisons between cases and controls regarding demographic profile was done using univariate logistic regression generating crude ORs (for age, as continuous variable) and chi 2 test (for gender), respectively. For cases, comparisons between gastrointestinal diagnoses and age categories were done using univariate logistic regression generating crude ORs. 

#### 2.5.2. Questionnaire Data Including Previous Medical History, Anthelmintic Routines and Pasture Management

Comparisons between cases and controls regarding previous medical history (colic, veterinary treated colic, diarrhea, weight loss), anthelmintic routines (last drug used, time since last deworming, farm deworming routine, prevalence of previous parasitic infection) and management factors (use of co-grazing, fecal removal routines in pasture/paddock, time at pasture/in paddock, stocking density) were made using univariate logistic regression generating crude ORs.

#### 2.5.3. Parasitological Status

The parasitological status of each horse was determined for cyathostomins as FECs, and for *S. vulgaris* PCR, *S. vulgaris* ELISA and *A. perfoliata* as positive/not positive. Strongyle egg counts were clustered into the following EPG groups: low egg shedders (EPG ≤ 200), medium egg shedders (250–500 EPG) and high egg shedders (>500 EPG) [33]. The difference in distribution of the EPG categories for each determinant, respectively, was investigated using univariate ordered logistic regression generating crude ORs. The *S. vulgaris* ELISA was identified as positive using a cut-off value of 13.47% of positive control according to Andersen et al. [32]. The number of positive samples (i.e., above the positive threshold) were presented as total number and proportion of the total. For *S. vulgaris* PCR, *S. vulgaris* ELISA, and *A. perfoliata*, the association between each parasitological outcome and various determinants (season, case/control, age, previous medical history, anthelmintic routines, gastrointestinal diagnosis and pasture management factors) was evaluated in separate logistic regression models (one per each specific combination of exposure and outcome). Additional analysis using the Wilcoxon matched-pairs signed-rank test for *S. vulgaris* ELISA was performed as distribution plots (using the original continuous data) for cases and controls, respectively, for each gastrointestinal disease category as well as for all cases compared with all controls. 

#### 2.5.4. Multivariable Logistic Regression Analysis

Variables with a univariable *p* value < 0.2 were considered for subsequent inclusion in a multivariable model. To avoid problems associated with collinearity, where variables were considered to be measuring the same exposure or were shown to be highly correlated (Pearson correlation coefficient > 0.8), the most statistically significant or biologically plausible variable was selected for inclusion in the model. Five submodels were initially created for the following dependent variables: cases/controls, strongyle fecal egg counts, *S. vulgaris* PCR, *S. vulgaris* ELISA, *A. perfoliata*. For each model, all variables with *p* < 0.20 from the univariate analysis with the specific dependent variable were added to create the full model. From each model, backward selection of variables was performed: the variable with the highest p-value was removed, one at a time, until all variables remaining in the model had *p* < 0.05. 

## 3. Results

### 3.1. Data on Horses

A total of 137 cases and 137 controls were included in the study. A total of 66.7% of all presenting gastrointestinal patients at the clinic during the period February 2017 to February 2018 were included in the study. The mean and median ages of the cases and control horses were 10.8 (± SD 5.8) and 11 years and 10.2 (± SD 6.0) and 9 years, respectively, with a range of one to 25 years in both groups. The control population consisted of 74 geldings (54.0%), 58 mares (42.3%) and five stallions (3.6%) and the case population of 62 geldings (45.3%), 65 mares (47.4%) and 10 stallions (7.3%). There were no statistical differences in either age (*p* = 0.53) or gender distribution (*p* = 0.83) between cases and controls. The control horses were represented by 23 different breeds, the majority of which were Warmbloods (54.7%), followed by pony breeds (16.8%), Icelandic Horses (7.3%), Thoroughbreds (5.8%), draft breeds (4.4%), and Standardbreds (1.5%), with 9.5% consisting of various other breeds. Similarly, cases, consisting of 26 different breeds, were predominantly Warmbloods (40.1%), followed by Icelandic Horses (15.3%), pony breeds (13.7%), Standardbreds (7.2%), draft breeds (6.6%) and Thoroughbreds (4.4%), with 10.9% consisting of various other breeds. 

### 3.2. Colic Diagnoses

The diagnoses of cases included impaction colic (large colon, small colon and cecal impactions) (38.0%), acute colitis (10.2%), large intestinal displacement (9.5%), peritonitis (8.0%), gas distention colic (5.8%), eosinophilic enteritis/colitis (3.6%), chronic colitis (2.2%), and other (7.3%). No specific diagnosis was reached in 15.3% of the cases (Table 2). Age had no association with type of diagnosis, with impaction colic common in all ages (*p* = 0.32). Of the horses diagnosed with peritonitis, the underlying etiology was known in three cases; esophageal rupture (n = 1) and non-strangulating intestinal infarction (n = 2). The remainder were classified as idiopathic. Abdominal surgery was performed in 12/137 cases (8.0%). Of the horses that had surgery performed, the final diagnoses were as follows: large colon displacement (4), ileocecal invagination (2), equine eosinophilic gastroenteritis/enterocolitis (2), non-strangulating intestinal infarction (2), gas distention colic (1) and undiagnosed (1). A total of four (2.9%) cases were euthanised, all of which had had abdominal surgery. Of the horses that had abdominal surgery and/or were euthanised, parasitic injuries were found in two cases, both of which were diagnosed with *S. vulgaris*-associated non-strangulating intestinal infarction of the colon. Both cases of ileocecal invagination had abdominal surgery without parasitic findings, e.g., *A. perfoliata*. The final diagnoses and number of cases in each group are presented in Table 2.

### 3.3. Assocation with Season

A total of 47 horses presented with colic during the winter months (December–February), equivalent to a third (34.3%) of the total number of cases. The most common diagnoses made during this season included different types of impaction colic (31.9%), intestinal displacement (17%), acute colitis (12.8%), peritonitis (10.6%), and equine eosinophilic gastroenteritis/enterocolitis (6.4%). In spring (March–May), 37 cases were included (27% of total caseload), the majority of which were impaction colic (37.8%), peritonitis (10.8%) and intestinal displacement (8.1%). Fewer horses with gastrointestinal disease presented during the summer, with 21 cases (15.3% of total caseload) during the months June–August. Furthermore, during the summer, impaction colic (large intestinal) was most commonly diagnosed (38%). Peritonitis, gas distention colic and acute colitis were each diagnosed in 9.5% of the cases during these three months. Twenty-three percent of the total number of cases presented during the autumn (September–November). Again, impactions were common and were diagnosed in almost half of the colic cases during this time period (46.9%). The second and third most common diagnoses during these months were acute colitis (15.6%) and gas distention colic (12.5%), respectively. No peritonitis cases were diagnosed in the autumn. There was no statistical association with season and the type of colic diagnosed (*p* = 0.32). 

### 3.4. Questionnaire Data

The owner of each horse participating in the study completed a questionnaire on previous colic history, deworming routines and pasture management (Table 1). The overall questionnaire response rate for cases was 92% (126/137) and for control horses, 98% (134/137); however, within the questionnaire, the response rate varied depending on the questions asked (74.5%–98%). The results from the questionnaire are presented in Table 3, Table 4 and Table 5.

### 3.5. Previous Medical History

Case horses had been owned by the present owner for a mean of 4.3 years (range of one month to 25 years), and control horses had been in the present ownership for a mean of 4.9 years (range of one month to 25 years). Of the cases, 50.4% (60/119) had previously reported episodes of colic, with an average of 2.2 colic episodes prior to the study (range 1–12). The reported prevalence of previous colic was 23.9% for the controls (32/134), with an average of 1.8 previous colic episodes (range 1–10). Cases were significantly more likely to have had previous episodes of colic, compared to controls (*p* < 0.01) (Table 3). A third of the cases (33.3%) had been attended by a veterinarian due to colic within the previous 24 months, which was significantly more often compared with control horses (8.0%) (*p* < 0.01). In addition, cases were more likely to have had diarrhea within the previous two months compared to controls, although with a conservative Bonferroni correction approach, this difference did not reach statistical significance (*p* = 0.03).

### 3.6. Anthelmintic Routines

Deworming history was known in 88.9% (112/126) of cases and 98.5% (132/134) of controls that responded to the questionnaire, although the overall response rate to individual anthelmintic routine questions again varied (74.5%–98.5%). Anthelmintic treatment was performed at various times prior to presentation: 0–3 months (18.9%), 3–6 months (25.4%), 6–12 months (31.1%) and over 12 months (24.6%). There was no significant difference in time from deworming between cases and controls (*p* = 0.46) (Table 4). Macrocyclic lactones (ML) was the anthelmintic group most commonly used (59%) at the last treatment occasion. In detail, 41.7% had treated with ivermectin (IVM), 3.3% with moxidectin (MOX), 14.2% with a combination of IVM/moxidectin and praziquantel (COMB), 5.4% with pyrantel (PYR) and 3.3% had used fenbendazole (FBZ). Cases and controls showed a tendency to differ in terms of which drug was used at the last treatment, with control horses more often treated with IVM or COMB, and owners of cases, more often unable to recall the drug used (*p* = 0.02) (Table 4). Overall, 32.1% of the owners did not remember which drug they had used on the most recent treatment occasion (cases 43.8%, controls 24.2%). 

In this data set, 28.8% of the horses were dewormed routinely without parasitic diagnostics being performed, where 12.3% of the horses were dewormed once per year and 16.5% of the horses 2–4 times per year. Selective treatment based on FECs was used by 35.8% of the owners, and 31.9% of owners used selective treatment based on both FECs and larval culture. Overall, 16% performed routine diagnostics for *A. perfoliata*, but this was more common within the control group (18.7%) compared to cases (13.5%). In addition, routine deworming 2–4 times per year was more often employed by owners of control horses (20.2%) compared to cases (12.7%). However, there were no significant differences between cases and controls in fecal diagnostics and deworming routines (*p*= 0.11–0.99) (Table 4).

One third of all horses were reported to have been infected with cyathostomins within the previous 24 months (cases 29.4%, controls 36.8%). In contrast, only 4.6% of horses were known to have been positive for *S. vulgaris* during the same time period. Case horses showed a higher prevalence of previous *S. vulgaris* infection (6.9% versus 2.6%), although this difference did not reach statistical significance (*p* = 0.10). Infection with *A. perfoliata* and *Parascaris* spp. were rarely reported to have occurred within the previous 24 months, and the data showed same frequency for both parasites (1.4% (1.0% cases, 1.7% controls)), see Table 4. However, previous parasitic infection history was unknown in the majority of horses (59.4% (61.8% cases, 57.3% controls)). Overall, previous parasitic infection was not significantly different between cases and controls (*p* = 0.10–0.79). 

### 3.7. Pasture Management

Regarding pasture management, it was rare for owners to report the use of mixed or alternate grazing with other species (9.8% (cases 10.6%, controls 9.0%)). However, the majority of owners employed the use of separate winter and summer paddocks/grazing areas (64.0% (67.5% cases, 60.9% controls)) (Table 5). Fecal removal was utilised overall by 46.2% of horse owners, with no statistical difference between cases and controls (54.6% and 45.9%, respectively) (*p* = 0.17). However, only a small proportion of owners declared that they removed feces at least twice weekly (7.1%), with almost two thirds of owners (65.3%) reporting that they removed feces less often than once weekly.

All horses had access to pasture during the summer months, with the majority (54.7%) kept at pasture 24 h per day during the summer, with no significant difference in time at pasture during the summer found between cases and controls (*p* = 0.96). Similarly, there were no differences found between the two groups in the number of horses kept solely outdoors in the winter (*p* = 0.88). Cases were more likely to be stabled > 12 h per day in the winter, compared to controls (65% compared to 55.6%), although no statistical difference was found (*p* = 0.13). 

Stocking density (hectare/horse) calculated from data regarding paddock size and number of horses/paddocks, was similar between cases and controls in both winter and summer (*p* = 0.22–0.68) (Table 5). The majority of horses, both cases and controls, were kept at stocking densities providing < 0.4 hectare/horse (72.8% winter, 64.7% summer). Overall, season (winter/summer) showed a significant difference in both time at pasture and stocking density, with more horses stabled > 12 h/d and kept at stocking densities providing less than 0.4 hectare/horse during the winter as compared to the summer (*p* < 0.01). 

### 3.8. Multivariate Model for Cases and Controls

The multivariate logistic regression model investigating factors associated with being a case resulted in a final model that included three variables: age, treated for colic during last 24 months and stocking density. There was a significantly higher odds of being a case if treated for colic within the last 24 months (OR 7.87 (95% C.I. 3.04–20.34), *p* < 0.001), when adjusting for stocking density and age.

### 3.9. Strongyle Fecal Egg Counts

The overall prevalence of strongyle eggs was 44.9% (123/274) and the prevalence was similar between cases and controls, 47.4% and 42.3%, respectively. The majority, 67.5% (185/274), of the horses were classified as low egg shedders (EPG ≤ 200), with 16.4% denoted as high shedders (>500 EPG). The mean strongyle EPG was 238 ± SD 397 in the case group of which 55.5% (76/137) of the horses were shedding ≤ 50 EPG. The mean strongyle EPG was 320 ± SD 688 in the control group of which 52.6% (72/137) of the horses were shedding ≤ 50. Both the case group and the control group revealed similar number of high egg shedding horses (>500 EPG), 18.2% (25/137) and 14.6% (20/137), respectively. No statistical difference in the number of low, medium and high shedding horses between cases and controls was found (*p* = 0.92). In the study population as a whole, age was associated with egg shedding, with increasing age resulting in an increase in the number of low shedding horses and a decrease in the number of high shedders (*p* < 0.01). No statistically significant association with season and EPG level was found (*p* = 0.38). 

Horses with a previous history of weight loss or diarrhea had higher average EPG levels (426 ± SD 778 and 400 ± SD 530, respectively) compared to horses without such complaints (268 ± SD 732 and 262 ± SD 544, respectively); however, these differences were not significant (*p* = 0.39–0.49). EPG levels were not associated with gastrointestinal disease in general (*p* = 0.92) (Table 6), nor with a particular colic diagnosis (*p* = 0.61). 

EPG levels were lowest in horses most recently dewormed (0–3 months), with time since deworming overall approaching a significant association with EPG levels (*p* = 0.01). However, previously used anthelmintic drugs were not associated with egg shedding (*p* = 0.22). Although horses whose owners employed the use of FECs, as opposed to regular deworming at least once/year, had higher mean EPG levels, anthelmintic routine had no significant association with EPG level (*p* = 0.06). The result of the previous fecal sample had no statistical relationship with EPG level (*p* = 0.50). 

Horses kept in smaller enclosures during the winter, often on their own, were more likely to be low shedders, compared with horses kept in larger paddocks but with more companions, although no statistical significance was shown (*p* = 0.01). No such trend was observed during the summer (*p* = 0.22). Time at pasture, either in the winter or summer, did not influence EPG levels (*p* = 0.07–0.49). 

### 3.10. Strongylus Vulgaris PCR 

Seven horses in the case group and eight horses in the control group were PCR positive for *S. vulgaris*. The overall occurrence of *S. vulgaris* in this study was 5.5% (15/274). Seven of these 15 horses (46.7%) were defined as low egg shedders with EPG ≤ 200. Age showed no significant association with the number of positive horses (*p* = 0.46). Fewer positive horses were demonstrated during the winter months, compared to the other seasons, although no statistical association with season was shown (*p* = 0.05).

The majority, 80% (12/15), of the *S. vulgaris*-positive horses were dewormed either between 6–12 months or ≥ 12 months prior to fecal sampling, with almost half (7/15, 46.7%) of the positive horses having been dewormed with an unknown anthelmintic product > 12 months ago. Two horses were dewormed with a macrocyclic lactone (ML) within 3–6 months but were still positive for *S. vulgaris*. In addition, four horses were dewormed either once (3/15) or 2–4 times (1/15) a year but were also positive for *S. vulgaris*. None of the horses that were positive for *S. vulgaris* had reported a positive previous fecal sample for their last sample. 

In the *S. vulgaris* PCR positive group, only 20.0% (3/15) of owners (compared to 31.9% overall in the study sample) utilised larval culture at least once yearly, with more horses (46.0%) regularly diagnosed by only performing FECs as compared to the study population as a whole (35.8%). Likewise, horses regularly dewormed once per year were more likely to be positive for *S. vulgaris* compared to horses that received anthelmintic treatment 2–4 times per year and to horses, where specific diagnostics for *S. vulgaris* were used. The majority of owners that performed diagnostics for *A. perfoliata*, also screened for *S. vulgaris* (78.6%). However, the number of horses positive for *S. vulgaris* in each group was low, making true comparisons difficult and no statistically significant association with anthelmintic routine and the number of positive horses was found (*p* = 0.32). There was no association between a positive PCR and gastrointestinal disease (*p* = 0.80) (Table 6) or with a specific diagnosis (*p* = 0.76).

Neither stocking density nor hours at pasture had a significant association with the number of *S. vulgaris*-positive horses (*p* = 0.42–0.80). 

### 3.11. Strongylus vulgaris ELISA 

The total *S. vulgaris* seroprevalence in this study was 62.2% (161/259). Serum was unavailable in 15 cases. No statistical differences between cases and controls in a number of horses positive in the ELISA test was shown (*p* = 0.83) (Table 6). However, when the separate gastrointestinal diagnoses were compared to controls matched by time of presentation, horses diagnosed with peritonitis showed significantly higher ELISA scores, expressed as a percentage of positive control, than their matched controls (*p* < 0.02) (Figure 1). 

Overall, there was a trend towards a greater prevalence of positive ELISA tests in older horses (>20) compared to young individuals (≤2 years) (81.3% and 44.4%, respectively), but this difference did not reach significance (*p* = 0.04). Overall, age had no significant association with ELISA score (*p* = 0.37). A lower prevalence was observed during the spring (60%) as compared to the rest of the year (71.4%–81.7%), but overall there was no significant association with season on ELISA score (*p* = 0.17). 

Of the ELISA positive horses, 39.8% (64/161) were regularly diagnosed only performing FEC, 29.8% (48/161) of the horses performed FEC combined with *S. vulgaris* diagnostics, 16.1% (26/161) of the horses were dewormed 2–4 times/year and 8.7% (14/161) of the horses were dewormed on a yearly basis. Horses dewormed based on FECs without further diagnostics were more likely to have a positive ELISA score, compared to other anthelmintic routines, although using the conservative Bonferroni correction, statistical significance was not shown (*p* = 0.02). Neither the time from last anthelmintic treatment nor the drug used was associated with a positive ELISA result (*p* = 0.37–0.81). 

Stocking density in either season had no significant association with number of positive horses (*p* = 0.77–0.90). Time at pasture during the summer was shown to have a significant association with the outcome of the ELISA test (*p* < 0.01), but no such association was demonstrated during the winter (*p* = 0.51). 

### 3.12. Anoplocephala perfoliata 

A total of 15.7% (43/274) of the horses were positive for *A. perfoliata*. Seventeen horses (12.4%) were positive for *A. perfoliata* in the case group and 26 horses (19.0%) in the control group, with no significant difference between the two groups (*p* = 0.14). Age had no significant association with *A. perfoliata* prevalence (*p* = 0.12). Although the highest number of positive horses was found in the spring (44.2%), no statistically significant seasonal differences were observed (*p* = 0.05). 

No association between gastrointestinal disease and a positive *A. perfoliata* sample was demonstrated (*p* = 0.14) (Table 6). In addition, no specific gastrointestinal diagnosis was associated with a positive *A. perfoliata* sample (*p* = 0.96). Of the only two cases diagnosed with ileocecal invagination, fecal samples were negative for *A. perfoliata* and no tapeworms were found during abdominal surgery.

The time since last deworming was not associated with a positive test for *A. perfoliata* (*p* = 0.92). Three horses were positive for *A. perfoliata* despite treatment with ML combined with praziquantel within the last 3–6 months and for one horse, within 0–3 months. Another horse had been treated with this combination within 6–12 months of fecal sampling. Previous anthelmintic drug use had no overall significant association with the occurrence of *A. perfoliata* (*p* = 0.06). Eight of the 43 positive horses (18.6%) were normally analysed for *A. perfoliata* on a yearly basis. The group with the highest proportion of individuals positive for *A. perfoliata*, were horses whose owners reported the use of specific diagnostics for tapeworm. However, no horse that had a previous sample positive for *A. perfoliata*, had a positive fecal sample in the study. Overall, there was no significant association with anthelmintic routine used and a positive test for *A. perfoliata* (*p* = 0.20). 

Although horses with at least 12 h of pasture/paddock access, regardless of time of year, had a higher prevalence of *A. perfoliata*, no statistically significant association with hours at pasture was shown (*p* = 0.04–0.06). The majority of positive horses were kept in paddocks allowing less than 0.4 hectares/horse, but no significant association with stocking density on *A. perfoliata* prevalence was demonstrated (*p* = 0.10–0.67).

### 3.13. Multivariate Model for Parasite Outcome

The multivariate ordered logistic regression model investigating factors associated with the fecal egg count in categories of low, medium and high egg shedders, resulted in a final model that included four variables: season, stocking density, any previous history of colic and time since deworming (*p* < 0.05 for all). There was an increased odds of a higher fecal egg category when time since deworming increased, with a 7.3 (95% C.I. 2.2 –24.1; *p* = 0.001) times increase in risk found when a horse had received anthelmintic treatment 6–12 months prior to presentation, compared to being treated within 0–3 months, when adjusting for season, stocking density in the winter and known history of previous colic. 

When investigated factors associated with a positive PCR result for *S. vulgaris,* the multivariate ordered logistic regression model resulted in one variable: time since deworming. Deworming more than six months prior to sampling, as compared to treatment less than six months before presentation, resulted in an increased likelihood of a positive *S. vulgaris* PCR of 4.4 (95% CI 1.20–15.80; *p* = 0.025). 

A multivariate ordered logistic regression for a positive *S. vulgaris* ELISA outcome resulted in a final model that included three variables: time in ownership, stabling > 12 h/d in the summer and deworming only after FECs. Deworming using only FECs was associated with an increased risk of a positive test, when adjusting for time of ownership and stabling > 12 h/d in the summer (OR 2.3 (95% C.I. 1.3–4.1; *p* = 0.025)). 

Factors investigated to be associated with presence of *A. perfoliata*, resulted in a final model that included two variables: season and stabled > 12 h/d in the winter. The odds for the presence of *A. perfoliata* in a fecal sample was 2.8 times higher (95% C.I. 1.2–6.4; *p* = 0.035) in March–May when adjusting for time stabled in the winter. 

## 4. Discussion

The present study demonstrated a high percentage of horses (cases and controls) were positive in the *S. vulgaris* ELISA test, amounting to almost two thirds of the horses included, indicating a high frequency of exposure to large strongyles in the population studied. The ELISA test assesses IgG (T) antibodies in *S. vulgaris* excretory proteins with a relatively high specificity (81%), and appears to reflect current infection or recent exposure, within the previous five months [31,34]. The number of positive horses in this study may be a surprisingly high figure, given the low individual *S. vulgaris* prevalence found using fecal PCR analyses (5.5%), but it is equivalent to the recently shown farm prevalence of *S. vulgaris* (61%) in Sweden [22]. 

Given the known pathogenicity of *S. vulgaris*, horses diagnosed with peritonitis were considered of particular interest to a possible *S. vulgaris* etiology. One case in this group was excluded due to the reason that peritonitis was known (esophageal perforation). However, two cases of peritonitis were confirmed on necropsy to have been caused by non-strangulating infarction secondary to *S. vulgaris* thrombosis. The remaining eight cases of peritonitis were classified as idiopathic and responded successfully to antimicrobial treatment, but none of these cases underwent exploratory laparotomy or necropsy. In addition to causing non-strangulating intestinal lesions, *S. vulgaris* has also been suggested as a possible cause of *Actinobacillus equuli* peritonitis, the most common bacterium isolated from horses presenting with idiopathic peritonitis [35]. The causal relationship, however, cannot be confirmed using fecal analyses, given that peritonitis associated with *S. vulgaris* would be due to the migrating larval stages and not the adult form of the parasite, and therefore before egg excretion is expected. None of the horses presenting with peritonitis in the present study were PCR positive for *S. vulgaris*. However, in the peritonitis group, the percentage of horses positive for the ELISA test was significantly higher (77.8%) than overall, with only two cases being negative (serum was missing in one case). In addition, horses diagnosed with peritonitis had significantly higher ELISA scores as compared to matched controls, which could be suggestive of an *S. vulgaris* etiology (*p* < 0.02). *S. vulgaris*-specific antibodies have been shown to be strongly associated with non-strangulating intestinal infarction, resulting in septic peritonitis; however, not with other types of colic [6,36]. The cause of eosinophilic intestinal lesions is still under debate, and previous studies do not convincingly demonstrate a parasitic etiology [37,38]. Nonetheless, it is interesting to observe that, although few cases were included, all but one case with an eosinophilic intestinal lesion were also positive in the ELISA test (80%, 4/5). In one case of eosinophilic enteritis, although no *S. vulgaris* larvae were found, thrombosed vessels were observed adjacent to the lesion. 

In agreement with other case–control studies, no significant differences in parasitological status could be demonstrated using coprological assays between horses presenting with gastrointestinal disease in general and controls [16,36,39]. Overall, the fecal prevalence of *S. vulgaris* as assessed by PCR analyses and the presence of cyathostomins and *A. perfoliata* eggs were either lower or comparable to previously published prevalences of these equine intestinal parasites, as outlined below. 

Although a possible role in colic incidence has been suggested [40], cyathostomins appear to be well tolerated without causing disease in most cases, apart from the well-known syndrome of larval cyathostominosis in young animals [12,13]. As in the present study, a recent publication showed no differences in strongyle EPG between non-surgical colic cases and controls [16]. Previous studies have, however, shown an association of cyathostomins with weight loss and diarrhea [12,15,41,42]. In the present study, although horses with a previous history of diarrhea or weight loss within the last two months had higher average EPG levels compared to the rest of the study population, there was considerable variation in EPG levels and this difference was not significant. In the case group, only two cases (2/14) of acute colitis and one case (1/3) of chronic colitis were high shedders (EPG > 500). There are, however, several causes of colitis in horses [43], and even in diarrhea caused by cyathostomins, fecal egg counts are often negative, with the L4 stage present in the feces instead [12,41]. In addition, it has been shown that neither egg nor larval counts correlate well with the horse´s actual worm burden [44].

The prevalence of *A. perfoliata* reported here is in accordance with a previous Swedish case–control study that showed positive samples in 13% (18/134) of horses [11]. Previous studies have shown conflicting results regarding the association of tapeworm infection and general colic [7]. However, *A*. *perfoliata* has consistently shown to be associated with ileocecal lesions [8,10,45,46,47,48,49]. The present study included few cases with confirmed ileocecal lesions (2/137) and was thus unable to demonstrate such a relationship. 

No differences in parasite control strategies between cases and controls were found in the present study. ML was the most commonly used anthelmintic group overall, in accordance with several previous publications from various countries [50,51,52,53,54,55]. However, within the study population as a whole, differences in anthelmintic routines were present, with 28.8% of the owners deworming their horses on a routine basis without prior parasitic diagnostics. Likewise, in a recently published study 21% of Swedish horse-owners employed routine anthelmintic treatment 1–4 times yearly, without using coproscopic analyses [22]. However, before the prescription-only legislation was implemented, only 1% of owners were reported to use fecal analyses [25]. National differences in anthelmintic routines were recently described, with Denmark showing a much greater use of FECs than Austria, Netherlands, Germany and the United States [19]. Interestingly, the frequencies of use of FECs were very low (1.8% and 3.1%, respectively) in Germany and Austria, although these countries have allowed only the use of prescription anthelmintic drugs since 1975. In Denmark, where the concept was introduced in 1999, the figure was 50% [19]. One reason for this discrepancy was suggested to be differences in administration of the law and perhaps how rigidly the legislation was adhered to [19]. Thus, although almost a third of the owners in the present study did not use fecal analyses, compared to the above countries, the current use of FECs in Sweden appears comparatively high and shows a clear increase since selective treatment was enforced. However, it has to be noted that only approximately one third of owners declared the use of specific diagnostics for *S. vulgaris,* which is concerning, given that, in the present study, the use of FECs alone to determine the need for anthelmintic treatment, without further diagnostics, was associated with an increased odds of horses being positive in the *S. vulgaris* ELISA test. 

Regular deworming, once or twice yearly, has been suggested to provide adequate control of *S. vulgaris* [21,56]. However, in the present study, although *S. vulgaris* was more common in horses dewormed six months or more prior to sampling, with time since deworming increasing the odds of a positive *S. vulgaris* sample, a total of six positive horses (40%) had either been dewormed within six months of sampling or were regularly dewormed at least once yearly. A post-mortem study in Sardinian horses demonstrated parasite-induced lesions in the cranial mesenteric arteries in all 46 horses in the study, and *S. vulgaris* larvae in 39%, despite treatment with broad-spectrum anthelmintic drugs at least three times per year [24]. Also, a recent study in Kentucky showed a high level of exposure to *S. vulgaris*, with 95/128 of horses being ELISA positive, despite treatment with anthelmintics on average twice yearly [57]. Considering clinical signs of disease may occur in as early as 2–4 months after infection [58], re-infection resulting in disease is possible despite regular anthelmintic treatment, and indicates that targeting the infected pasture is critical in management, and that eradication of *S. vulgaris* is unlikely using annual or bi-annual anthelmintic treatment, if horses are still allowed to graze in contaminated pastures. In addition, an experimental study in foals indicated that ivermectin may have little or no effect against migrating L5 larvae [59], suggesting that treatment may be less effective than often assumed. 

Similar to a previous study on parasite control strategies in Sweden [25], only a minority of owners used alternating grazing species (9.8%), one method to reduce pasture contamination. This is low compared to some countries, such as Ireland, where co-grazing was used by the majority of owners [55]. However, most owners in the present study employed the use of separate winter and summer grazing areas. The regular use of fecal removal from the pasture can be an effective method to reduce parasite burden [60,61] and may be as effective as regular anthelmintic treatment, if performed twice weekly [61]. In the present study, although almost half of the owners reported removing feces from the pasture or paddock, only a minority did so at least twice weekly. The effect of fecal removal done more seldom than once weekly, which was the case in almost two thirds (65.4%) of owners that removed feces, is most likely poor, given that infective larvae will develop from excreted eggs within 2–3 weeks in spring and fall, and in as short as three days under ideal conditions [62]. In addition, most horses, both cases and controls, were kept at high stocking densities, provided with < 0.4 hectare/horse both in summer and winter. A high stocking density, with a large number of horses in paddocks of limited space, increases the risk of parasitic infections [63], and, in general, 1–2 acres/horse (corresponding to 0.4–0.8 hectare) is recommended [64]. 

Given the prospective nature, the study population in the present study was dependent on whichever gastrointestinal cases presented to the hospital within the set time-frame and, gastrointestinal diagnoses of particular interest with regards to a parasitic etiology, such as non-strangulating intestinal infarctions and ileo-cecal lesions, were, unfortunately, few. In addition, the study included a relatively small number of severe and/or surgical cases, probably because such cases required more urgent attention, resulting in lower consideration for study participation. 

## 5. Conclusions

In conclusion, the present study demonstrated a high level of exposure of Swedish horses to *S. vulgaris*, as shown by the majority of horses, both cases and controls, being positive in the ELISA test. In addition, horses with peritonitis had significantly higher ELISA values as compared to controls and other gastrointestinal diagnoses, suggestive of an *S. vulgaris* etiology in this case group. Future studies addressing the association of *S. vulgaris* and peritonitis in the Swedish horse population is highly desirable, in order to further elucidate the clinical implication of the increasing prevalence of *S. vulgaris* in this country.

Furthermore, the present study revealed important information regarding the current use of anthelmintic treatment regimes in Sweden. For example, although the study showed a clear increase in the use of fecal analyses compared with before the prescription-only law was implemented, such legislation does not, per se, result in all owners using fecal diagnostics prior to treatment. The use of specific diagnostics is still low, indicating a need for education, of both owners and the veterinary profession, on how to best apply the diagnostic tools available. Treating horses annually or bi-annually, did not ensure a negative test result for *S. vulgaris*. Moreover, the study revealed that stocking intensity is often high and practices such as frequent fecal removal or co-grazing with other species were rarely performed. Thus, in addition, educating owners regarding optimal pasture management appears vital. In the advent of the increasing resistance to the anthelmintic drugs available together with the increase in prevalence of *S. vulgaris*, strategies for sustainable control of equine internal parasites are urgently needed, including pasture management.

## Figures and Tables

**Figure 1 animals-10-00638-f001:**
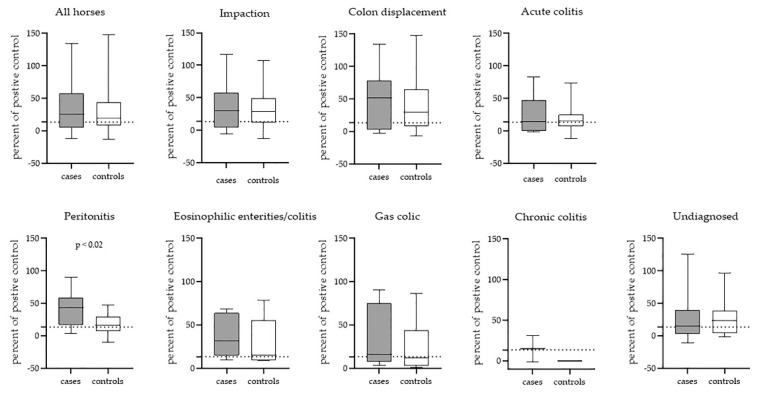
A boxplot illustrating the quantitative enzyme-linked immunosorbent assay (ELISA) values for each colic diagnosis compared with their matched controls. The bold line indicates median, the dotted line indicates the cut-off (13.47) value for positive ELISA.

**Table 1 animals-10-00638-t001:** Questionnaire data collected from participating cases and controls.

Information	Descriptor
Age of horse	Years
Time in ownership	Years
Any previous history of colic ^a^	Yes/No
If yes, specify number of colic episodes	Number
Treated for colic by a veterinarian during last 24 months	Yes/No
Signs of diarrhea ^b^ during last 2 months	Yes/No
Lost weight ^c^ during last 2 months	Yes/No
Last anthelmintic treatment (months)	0–3; 3–6; 6–12; >12
Drug used during last treatment	FBZ ^d^; PYR ^e^; IVM ^f^; MOX ^g^; COMB ^h^
Deworming routines applied on the farm	(i) only after FEC ^i^ (ii) after FEC and *S. vulgaris* diagnostics (iii) diagnostics for *A. perfoliata* at least 1/year (iv) routine deworming once/year (v) routine deworming 2–4 times/year
Presence of parasites in fecal sample during last 24 months	Small strongyles; *S. vulgaris*; *A. perfoliata*; *Parascasris spp*; do not know
Access to separate winter and summer pasture	Yes/No
Grazing with other animals species	Yes/No
Size of pasture or paddock, winter	Hectare
Size of pasture or paddock, summer	Hectare
Number of horses in pasture/paddock, winter	Number
Number of horses in pasture/paddock, summer	Number
Use of fecal removal	Yes/No
If yes, how often are feces removed	Number/week
Hours spent outdoors (paddock), winter	1–24
Hours spent outdoors (pasture), summer	1–24

^a^ restlessness and pawing at the ground, irritated kicking to the stomach, rolling or attempting to roll; ^b^ loose consistency of feces; ^c^ as subjectively appreciated by owner; ^d^ fenbendazole; ^e^ pyrantel; ^f^ ivermectin; ^g^ moxidectin; ^h^ ivermectin/moxidectin and praziquantel; ^i^ fecal egg count.

**Table 2 animals-10-00638-t002:** Distribution of gastrointestinal diagnoses in cases.

Diagnosis	Number of Cases (% of Total Case Load)
Impaction (small colon, large colon, cecal)	52 (38.0%)
Acute colitis	14 (10.2%)
Large colon displacement	13 (9.5%)
Peritonitis	11 (8.0%)
Gas distention colic	8 (5.8%)
Equine eosinophilic gastroenteritis/enterocolitis (EEG)	5 (3.6%)
Chronic colitis	3 (2.2%)
Undiagnosed	21 (15.3%)
Other	10 (7.3%)

**Table 3 animals-10-00638-t003:** Results of questionnaire data: owner-reported previous medical history of cases and controls as outlined in a questionnaire presented after agreement of study participation. The *p*-value for statistical significance was corrected for the number of analyses (n = 5) according to Bonferroni, i.e., *p* < 0.01 (0.05/5).

Information	Cases	Controls	OR (95% C.I.)	*p*-Value
Prevalence with previous history of colic ^a^ (%)	50.4%	23.9%	3.13 (1.83–5.35)	<0.01
Number of previous colic episodes (mean ± SD)	2.2 (± 2.3)	1.8 (± 1.9)	1.09 (0.86–1.39)	0.47
Prevalence treated for colic during last 24 months (%)	33.3%	8%	5.73 (2.78–11.84)	<0.01
Prevalence with diarrhea ^b^ during last 2 months	15.8%	7.5%	0.42 (0.18–0.94)	0.03
Prevalence with weight loss ^c^ during last 2 months	9.2%	6.0%	0.45 (0.18–1.10)	0.08

^a^ restlessness and pawing at the ground, irritated kicking to the stomach, rolling or attempting to roll; ^b^ loose consistency of feces; ^c^ as subjectively appreciated by owner.

**Table 4 animals-10-00638-t004:** Result of questionnaire data: anthelmintic routines. The *p*-value for statistical significance was corrected for the number of analyses (n = 4) according to Bonferroni, i.e., *p* < 0.01 (0.05/4).

Information	Cases	Controls	*p*-Value
**Time from last helmintic treatment (months) (%)**			
0–3	17.9%	19.7%	0.38
3–6	28.6%	22.7%	0.40
6–12	26.8%	34.8%	0.66
>12	26.8%	22.7%	0.51
For the variable (all time groups) cases/controls:			0.46
**Drugs used at last treatment (%)**			
FBZ ^a^	4.5%	2.3%	0.48
PYR ^b^	3.6%	6.9%	0.16
MOX ^c^	3.6%	3.1%	0.62
IVM ^d^	35.5%	46.9%	0.21
COMB ^e^	10%	17.7%	0.13
Unknown	42.7%	23.1%	0.94
For the variable (all drug groups) cases/controls:			0.02
**Deworming routines applied on the farm (%)**			
(i) after FEC ^f^	35.7%	35.8%	0.99
(ii) after FEC and *S. vulgaris* diagnostics	30.2%	33.6%	0.55
(iii) diagnostics for *A. perfoliata* at least once/year	13.5%	18.7%	0.26
(iv) routine deworming once/year	13.5%	11.1%	0.57
(v) routine deworming 2–4 times/year	12.7%	20.1%	0.11
**Presence of parasites in fecal sample during last 24 months**			
Cyathostomins	29.4%	36.8%	0.13
*S. vulgaris*	6.9%	2.6%	0.10
*A. perfoliata*	1.0%	1.7%	0.79
*Parascaris* spp.	1.0%	1.7%	0.79
Do not know	61.8%	57.3%	0.31

^a^ fenbendazole; ^b^ pyrantel; ^c^ ivermectin; ^d^ moxidectin; ^e^ ivermectin/moxidectin and praziquantel; ^f^ fecal egg count.

**Table 5 animals-10-00638-t005:** Results of questionnaire data: pasture management. The *p*-value for statistical significance was corrected for the number of analyses (n = 10) according to Bonferroni, i.e., *p* < 0.005 (0.05/10).

Information>	Cases	Controls	*p*-Value
Access to separate summer/winter paddocks	67.5%	60.9%	0.27
Grazing with other animal species	10.6%	9.0%	0.67
Fecal removal Fecal removal from pasture, overall	54.6%	45.9%	0.17
Fecal removal from pasture, ≥2x/week	5.9%	8.1%	0.33
Time stabled/outdoors Outdoors 24 h/d, winter	14.2%	14.8%	0.88
Stabled > 12 h/d, winter	65%	55.6%	0.13
Outdoors 24h/d, summer	54.2	54.8%	0.96
Stabled > 12 h/d, summer	26.7%	26.7%	0.93
Stocking density Stocking density, summer (hectare/horse)	0.6	0.8	0.68
Stocking density, winter (hectare/horse)	0.4	0.4	0.22

**Table 6 animals-10-00638-t006:** Fecal and serology results in cases and controls. The *p*-value for statistical significance was corrected for the number of analyses (n = 4) according to Bonferroni, i.e., *p* < 0.01 (0.05/4).

Investigated Parameter	Cases n (%)	Controls n (%)	*p*-Value
EPG ^a^ level LOW	93 (67.9%)	92 (67.1%)	0.92
EPG ^a^ level MEDIUM	19 (13.9%)	25 (18.2%)	
EPG ^a^ level HIGH	25 (18.3%)	20 (14.6%)	
*S. vulgaris* PCR	7 (5.2%)	8 (5.8%)	0.80
*S. vulgaris* ELISA	75 (61.5%)	86 (62.8%)	0.83
*A. perfoliata*	17 (12.4%)	26 (19.0%)	0.14

^a^ EPG, eggs per gram of feces.

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
