# Peer review of "Parasite Occurrence and Parasite Management in Swedish Horses Presenting with Gastrointestinal Disease—A Case–Control Study"

_animals, 2020, doi:10.3390/ani10040638_

Round 1
Reviewer 1 Report
Overall the manuscript is well written and M&M clear and follow good scientific procedures.
I only have some minor comments:
L139 - convert 1" to SI units
L138,L141, a43 etc. "ml" is not written consistently.
L142 - correct format for centrifugal speed is: 2000 x g as written in L152
Table 3a (and elsewhere) p-values should not be truncated but quoted to 3 decimal places. p=0.000 is incorrect and should be written as p<0.001
Also p-values should be consistenlty written to 3 decimal places (see Table 3b, etc)
L 292 16%
Author Response
Dear Editor and reviewers,
Detailed answers are found in the attachment.
Best regards,
Eva

Reviewer 2 Report
This manuscript describes a case-control study investigating the association between colic and parasite levels. The presence of parasites was assessed using several diagnostic tests and each case and control was retrospectively asked about parasite management practices and other demographic characteristics.
In general, the manuscript is quite well-written. The authors provide a good outline to the study and explain the clinical procedures (e.g. testing and diagnostic criteria) in enough detail for the reader to follow. Given the number of ways the outcome could be assessed (ELISA, PCR, and FEC), it does get a bit confusing from the start of the statistical analysis section (line 171) following through the results. Perhaps it would be helpful creating sub-sections under the statistical analysis and results to describe the procedures (and results) per outcome/test? Otherwise, I do get a bit lost following which test was used for which outcome.
For the statistical analysis, the manuscript states that univariable logistic regression was used, but multivariable logistic regression would provide a better estimate of the association between variables. Going through the results, I would suspect that several independent variables might be correlated with each other (e.g. diarrhoea and weight loss) and confounding might be present -- adjusted odds ratios, through multivariable logistic regression, would provide a better indication on the relationships. Also, please provide confidence intervals when presenting odds ratios.
Lastly, please provide a discussion of any limitations or other considerations that the readers should be aware of -- for example, generalisations that can be made from this study. Given that the sample was taken from a referral hospital, the cases seen here might be more severe or complex than cases that might be seen in primary care.
Author Response
Dear Editor and reviwers,
Detailed answers are found in the attachment.
Best regards,
Eva Tydén

Round 2
Reviewer 2 Report
Thank you for responding to my comments. The revised manuscript is well-written and I have no further comments to add.